# Fatigue of an Aluminum Foam Sandwich Formed by Powder Metallurgy

**DOI:** 10.3390/ma16031226

**Published:** 2023-01-31

**Authors:** Sitian Liu, Peng Huang, Xi Sun, Wenqi Zeng, Jiatong Zhang, Guoyin Zu

**Affiliations:** School of Materials Science and Engineering, Northeastern University, Shenyang 110819, China

**Keywords:** aluminum foam sandwich (AFS), fatigue life, powder metallurgy, fatigue damage

## Abstract

In this paper, an aluminum foam sandwich (AFS) was prepared by the rolling composite-powder metallurgy method, and its fatigue properties were studied. It was compared with an AFS made by the adhesive method to study its fatigue properties more directly. In this experiment, the fatigue performance was investigated by studying the microscopic interface, fatigue life, deflection curve, and failure mode. The results show that the fatigue life of an AFS with the rolling composite-powder metallurgy method is much longer than that with the adhesive method. The failure mode of an AFS made by the rolling composite-powder metallurgy method is shear failure, and that of an AFS made by the adhesive method is shear failure and interface debonding. An AFS with the rolling composite-powder metallurgy method has better fatigue properties. This paper also explored the fatigue damage model using the fatigue modulus method, and the polynomial fitting method has a higher fitting degree.

## 1. Introduction

Aluminum foam is a new metal material. Because of its low density, high specific strength, sound absorption, energy absorption, and other performance, it has a wide range of applications in machinery, construction, transportation, electronics, and even aerospace [1,2,3,4,5]. An Aluminum foam sandwich (AFS) is a new composite material, which consists of dense panels on both sides and an aluminum foam core layer in the middle, forming the so-called “sandwich” structure. The AFS has an excellent performance of aluminum foam and high strength and hardness of the panel, so it has a very promising application [6].

When an AFS is used as a structural material, its static properties are firstly considered. Its bending properties [7,8,9], compression properties [10,11,12], and impact properties [13,14,15,16] were studied. In addition, in the practical application scenarios, the specimens are mostly subjected to the cyclic loading, so it is necessary to study the fatigue properties of the AFS. Compared to the fatigue properties of the AFS, the fatigue properties of aluminum foam have been studied firstly. Harte et al. [17] tested the fatigue properties of an open-cell aluminum foam and closed-cell aluminum foam with different compositions, and the results revealed that the fatigue ratio was independent of the average stress and the density of aluminum foam. Zettl et al. [18] explored the fatigue properties of an Al—Mg—Si foam metal and Al—Si foam metal, and the results indicated that the fatigue data had obvious dispersion, and there were always initial defects at the location of the crack initiation. Hu et al. [19] investigated the effect of large cells on the fatigue performance of a closed-cell aluminum foam by numerical simulations and experiments and found that the fatigue life of aluminum foam depended mainly on the proportion of large cells rather than on the distribution of cell sizes. Zhao et al. [20] explored the tensile fatigue performance of aluminum foam and obtained the probability—stress—life curve, which could effectively describe the dispersion of fatigue life.

Inspired by the research on the fatigue performance of aluminum foam, the fatigue performance of the AFS has been studied mainly in the following aspects: factors affecting fatigue life, deformation characteristic, fracture mode, and predicted life. Yao et al. [21] studied the fatigue performance of an AFS made by the liquid diffusion welding method; the results showed that compared with an AFS made of the adhesive method, the metallurgical bonding method had higher bonding strength and longer fatigue life, and the failure modes of the two methods were also discussed. Yao et al. [22] investigated the fatigue performance of an AFS with different densities, described the fatigue process by force cycle curves and deflection curves, and proposed a way to predict fatigue life. Harte et al. [23] investigated the fatigue performance of an AFS with four-point bending, determined the different failure modes caused by the size effect, obtained the failure mode diagram of the AFS, and verified the effectiveness of the diagram through experiments. Jen et al. [24] investigated the effects of temperature and moisture on the static and fatigue properties of aluminum foam sandwich structures, and the experimental results showed that moisture had little effect on static strength and fatigue strength, and temperature played a more important role. Yan et al. [25] explored the fatigue damage mechanism of carbon fiber panel structures and proposed three microscopic damage mechanisms: stress concentration at the cell ridge or cell wall, damage at the thinnest position of cell wall, and damage caused at the intersection of cell wall.

At present, scholars are more concerned with an AFS prepared by the adhesive method, while the fatigue properties of an AFS made by the rolling composite-powder metallurgy method (hereinafter referred to as the powder metallurgy method) have rarely been reported. Based on the preparative study by Sun et al. [26], this paper investigated the fatigue properties of an AFS made by the powder metallurgy method. The bonding interface was analyzed using optical microscopy, the fatigue properties were compared with AFS prepared by the adhesive method, and the fatigue fracture mode was investigated.

## 2. Materials and Methods

### 2.1. Preparation of Materials

In this paper, the AFS was prepared by the powder metallurgy method [26]. The specific operation process was as follows. The four metal powders of Al, Si, Mg, and Cu were evenly mixed with the blowing agent TiH_2_. The four metal powders and blowing agent TiH_2_ are common commercial elements, and detailed parameters are shown in Table 1. In order to improve the decomposition temperature of the blowing agent TiH_2_ and ensure that the decomposition temperature of TiH_2_ matched the melting temperature of the core powder, the TiH_2_ should be pre-oxidized. The temperature was 470 °C and the time was 1.5 h [27]. The mixed powder was sealed in the cavity of the aluminum alloy, which was made of the aluminum alloy 3003 with a thickness of 4 mm. To ensure that a good interfacial flatness was finally obtained, the cavity was cleaned and immersed in 40 g/L aqueous sodium hydroxide solution for 10–20 min and then in the dilute hydrochloric acid solution for 10 min to remove the impurities on the surface [26]. The sealed precursors conducted cold rolling and hot rolling to obtain the foaming precursors, respectively. The cold rolling process adopted the multi-pass small depression process. The rolling rate was 6 mm/min, and the core layer powder density was about 85% after cold rolling. The hot rolling temperature range was 330–400 °C, and the single pass depression rate was 20–30%. Then the foaming process was carried out and the foaming temperature was 620 °C for 10–15 min. In order to heat quickly during the foaming process, a layer of graphite with uniform thickness was coated on the surface of the precursors. The AFS could be obtained after cutting with a wire cutter. The process flow chart of the AFS prepared by the powder metallurgy method is shown in Figure 1. According to the above method, an AFS with a height of 24 mm (the thickness of the upper and lower panels is 2 mm) was prepared for the follow-up experiments.

In order to compare the fatigue performance more intuitively, this paper selected an AFS prepared by the adhesive method and the AFS made by the powder metallurgy method. The glue was epoxy resin adhesive E-44 and hardener 650, 1:1 evenly mixed; the panel was bonded with the aluminum foam core layer, then put into the drying box at 80 °C for 2 h and rested for 48 h [28].

### 2.2. Quasi-Static Three-Point Bending Experiment

Before the fatigue tests, the quasi-static three-point bending tests were conducted to obtain the ultimate bending load to guide the fatigue tests. The model AG-XPLUS100KN electronic universal testing machine produced by Shimadzu, Japan, was used with a speed of 2 mm/min, and a span length of 100 mm. The sample size of the quasi-static three-point bending tests was the same as that of the fatigue tests.

### 2.3. Fatigue Test

Referring to the ASTM C393 standard, the specimen size was 170 mm × 30 mm × 24 mm. A series of three-point bending fatigue tests were carried out with the fatigue testing machine produced by Shimadzu, Japan, Maximum load 20 KN. The fixture size requirements are shown in Table 2, and the position relationship between the three-point bending fixture and the sample is shown in Figure 2. *l* represented the span length, *d* represented the indenter diameter, *c* represented the core thickness, *t* was the panel thickness, and *S* was the sample length. The test was stress-controlled with a cycle frequency 15 Hz, and the constant amplitude sine loading was adopted. The load ratio was *R* = 0.1, and *R* was expressed as the ratio of the minimum load value *F*_min_ to the maximum load value *F*_max_, as per Equation (1). *F*_max_ and *F*_min_ represented the maximum and minimum load value, respectively. The fatigue load curve of the three-point bending test is shown in Figure 3. The load level was expressed as the ratio of the maximum load value to the ultimate bending load, as shown in Equation (2). $F_{P}$ represented the ultimate bending load in the quasi-static three-point bending test. In fatigue experiments, four load levels were generally set, respectively, 95%, 90%, 80%, and 70%. All experiments were carried out at room temperature, and at least four valid data were obtained from the AFSs of different preparation methods. The deformation process was recorded using a digital camera, and the displacement values under different cycles were recorded. The test ending condition was that there were obvious cracks, and the middle deflection value of the upper panel was 15% or more of the thickness.
(1)$$R=\frac{F_{min}}{F_{max}}$$
(2)$$r=\frac{F_{max}}{F_{P}}$$

### 2.4. Sample Preparation of Microstructure

To systematically analyze the fatigue properties of the AFSs under different preparation methods, the microstructure of the bonding interface of different preparation methods was obtained. Sample preparation steps were as follows. The sample size was 20 mm × 20 mm × 15 mm. Four kinds of sandpaper of 800, 1200, 1500, and 2000 mesh were selected to sand the surface of the sample until there were no obvious scratches on the surface, and then polished until the surface was bright. The surface was cleaned with alcohol, dried with a hairdryer, and observed with an optical microscope. In this paper, the Olympus DSX500 optical digital microscope was used for microscopic observation.

## 3. Results and Discussion

### 3.1. Microstructure

The interface microstructure of the AFS is shown in Figure 4. As shown in Figure 4a, it can be seen that the bright and dark intersection marked by the red dotted line was the metallurgical bonding interface of the AFS made by the powder metallurgy method. The dark area was the aluminum alloy panel, the bright area was the core layer area, and the boundary between the bright and dark area was a straight line with a flat interface and no obvious defects. Huang et al. [27] explored the microstructure of an AFS made by the powder metallurgy method and found the same metal elements at the bonding interface and near the cell wall. Si, Cu, and Mg elements of the core layer diffused to the metal panel. It was estimated that the metallurgical bonding interface was about 25 μm. The XRD analysis results of the cell wall showed that the main phases were Al, Si, Al_2_Cu, Mg_2_Si, and Al_4_Cu_2_Mg_8_Si_7_. The EDS analysis results of the metallurgical bonding interface showed that the AlSi solid solution phase, the Al_2_Cu phase, and the Al_4_Cu_2_Mg_8_Si_7_ phase were the main phases in the metallurgical bonding interface. The same phase was found at the metallurgical bonding interface and cell wall, indicating that elements in the core layer had spread to the metallurgical bonding interface. Based on the above results, it can be shown that the good metallurgical bonding interface is formed at the bonding interface and the interface is flat without obvious defects. From Figure 4b, it can be seen that the red dotted line was where the adhesive interface combined, and the area between the aluminum panel and the foam core layer was filled with glue. The interface was not flat and more uneven with obvious defects. In addition, there were many impurities, and these defects and impurities might be the place where cracks were easy to form in the fatigue process. Additionally, it can be seen that the glue layer had many irregular cells, which was because the glue mixed with air in the solidification process, causing many cells in the interface, and these affected the performance of AFS. In conclusion, the metallurgical interface of the AFS made by the powder metallurgy method is flat and can bear a higher load.

### 3.2. Quasi-Static Test Result

The load-displacement curves of the AFSs made by two different preparation methods under the quasi-static three-point bending load are shown in Figure 5. It can be concluded that the load-displacement curve could be roughly divided into three stages. The first stage was the linear elastic stage, the load grew rapidly in a short time, and the load and displacement had a linear increasing relationship. If the load was removed during this stage, the specimen could return to the original form. The slope of the straight line was approximately the sample’s stiffness. At this stage, the main force was born by the panel, and there was no change in the core layer. The second stage was the platform stage, where the specimen’s load gradually decreased to a certain load value and stabilized after reaching the ultimate load, and the measured load value of the platform stage was 60% of the ultimate load value. In this stage, the panel near the indenter was the first to buckle, then the core layer cracked and gradually expanded, and the sandwich panel began to deform. The third stage was the failure stage, where the core layer cracks gradually expanded under the load. When the applied load value reached the core layer’s shear load, the cracks expanded from near the upper indenter to the lower surface, and the specimen was the shear failure. From the figure, it can be seen that the ultimate bending load of the AFS made by the powder metallurgy method is 4909 N, while that of the AFS made by the adhesive method is 4261 N. The ultimate bending load of the powder metallurgy method is greater than that of the adhesive method. The results show that the AFS made by the powder metallurgy method has stronger bending resistance than that made by the adhesive method.

### 3.3. S-N Curve

Three-point bending fatigue tests were conducted on the AFSs prepared by the powder metallurgy method and by the adhesive method, respectively. The results are shown in Figure 6. It can be seen from the figure that when the load level is 90%, the fatigue life of the powder metallurgy method and the adhesive method is 148 and 77 cycles, respectively; when the load level is 70%, the fatigue life of the powder metallurgy method and the adhesive method is 789,508 and 615,201 cycles, respectively. The specific values of the fatigue life are shown in Table 3. At the same load level, the fatigue life of the powder metallurgy method was higher than that of the adhesive method. Another point to note was that when the load level value was larger, that was when the fatigue specimen was subjected to the higher load, the fatigue life’s difference between the powder metallurgy method and the adhesive method was not significant. When the load level was lower, and the specimen applied the smaller fatigue load, the fatigue life’s difference between the two methods was very obvious. Therefore, we can conclude that the overall fatigue life of the powder metallurgy method is better than that of the adhesive method. The smaller the fatigue load on the specimen, the more obvious this superiority.

It could be easily seen from Figure 6 that the load level and the logarithm of fatigue life were approximately linear, so the *F*_max_/*F*_P_-lg*n* curves were fitted and the fitting straight lines correspond to the equation as:(3)$$\frac{F_{max}}{F_{P}}=a+blgn$$
where *F*_max_ indicates the maximum load in the fatigue test, *F*_P_ indicates the ultimate bending load measured in the quasi-static three-point bending test, *n* indicates the number of fatigue cycles, and *a* and *b* are constants.

The exponential function was introduced on both sides of Equation (3) and collated to:(4)$$n=10^{\left(cr-d\right)}$$
where $r=\frac{F_{max}}{F_{P}}$, *c* and *d* are constants. This equation can predict the fatigue life of the AFS, and the fitting curves are shown in Figure 6. This function could fit the present experimental data approximately. To sum up, it can be concluded that the fatigue life of the powder metallurgy method is higher than that of the adhesive method for the same load level, and Formula (3) can predict the fatigue life very well.

### 3.4. Deflection Curve

In order to discuss the mechanical variation behavior of the AFS under the fatigue load in more detail, the variation curves of deflection with fatigue life under different load levels were obtained, as shown in Figure 7. Deflection indicates the deflection value under each cycle. From Figure 7a, it could be seen that the Deflection-lg*n* curves at different load levels had the same variation trend, and the fatigue process was roughly divided into two stages: the stable stage and the transient breakage stage. The stable stage accounted for about 90% of the whole fatigue process. In the stable stage, the displacement changed slowly with the cyclic change of the fatigue load, and the specimen was not damaged and had no macroscopic cracks. However, with the increase of the fatigue cycle number, the fatigue damage accumulated and expanded inside the specimen. When the fatigue damage accumulated to the critical collapsing level, macroscopic cracks appeared in the specimen, and the cracks expanded rapidly and penetrated the whole specimen. The fatigue fracture occurred on the specimen. Figure 7b had the same variation trend as Figure 7a. The length of the stabilization stage could approximately represent the fatigue life of the specimen. Comparing Figure 7a,b, the length of the stabilization stage in Figure 7a is longer than in Figure 7b, which indicates that the fatigue life of the powder metallurgy method is better than that of the adhesive method.

The curves of relative displacement *d*_0max_/*d*_max_ changing with the fatigue cycle number at different load levels are shown in Figure 8. The *d*_0max_ indicates the maximum displacement value at the first cycle, and the *d*_max_ indicates the maximum displacement value at each cycle. The trend of this figure was the same as the previous one, and the slight difference with Figure 7 was that this figure could reflect a mechanical behavior change in the whole pre-stage more accurately. At the beginning of the fatigue process, the relative displacement decreased rapidly, indicating that the specimen had already been damaged and caused certain damage degree at this time. This stage accounted for about 30% of the whole fatigue stage. The second stage was the stable stage, the relative displacement remained stable, and the specimen had no macroscopic cracks. This stage accounted for about 60% of the whole fatigue stage. The third stage was the transient fracture stage, when the fatigue damage accumulated to a critical collapsing level, the specimen occurred fail and fracture. It can be noted that in Figure 8a, the relative displacement *d*_0max_/*d*_max_ value of the load level of 80% is slightly higher than that of the load level of 70%. This is because there were large cells in the AFS that had not been observed. If the indenter was near the large cells, the displacement value would increase, finally the relative displacement value changed. This phenomenon has also been found in other studies [21], which did not affect fatigue life. Comparing Figure 8a,b, the stabilization stage in Figure 8a is longer, indicating that the specimens made by the powder metallurgy method are more stable than those made by the adhesive method.

### 3.5. Failure Modes

Harte et al. [23] analyzed the fatigue failure modes of sandwich structures and classified the fatigue failure modes into three categories, which were face fatigue, core shear, and indentation fatigue. Face fatigue referred to when the panel of the sandwich structure was a low yield strength material; the ultimate yield strength of the panel determined the overall ultimate strength of the sandwich structure, and the failure of the sandwich panel specifically showed that the face could fracture before the core. Core shear meant that when the sandwich structure was subjected to the bending load, the transverse shear force mainly acted on the core layer, which eventually caused the shear failure of the core layer. Indentation fatigue referred to the formation of three plastic hinges around the panel in the vicinity of the upper indenter, and the core layer was compressed by the surface. Harte also derived the ultimate load equations for the relevant failure modes.

Based on the analysis and study above, the macroscopic morphology of the fatigue fracture of AFSs with two different preparation methods is shown in Figure 9. The places where the yellow dotted lines were drawn in Figure 9a,b were cracks. It can be seen that the fracture mode of the powder metallurgy method is core shear, and the first place where the cracks appeared may be at the large cells or may be near the upper indenter. The large cells were a solidification defect, and this location was the first place to occur stress concentration. When the load accumulated to a critical collapsing level, the macroscopic cracks occurred at the large cells. Because the upper panel was subject to compressive stress and the lower panel was subject to tensile stress, the crack extended in the 45° direction toward the panel, and finally showed fatigue failure of the sample. The fracture mode of the adhesive method is core shear and interface debonding. When the crack extended to the bonding interface, the panel debonded with the core layer. The powder metallurgy method does not show this phenomenon, which can also reflect that the bonding interface strength of the powder metallurgy method is higher than that of the adhesive method.

## 4. Fatigue Damage

### 4.1. Fatigue Modulus

In the above paper, we proposed the formula for fitting the load levels and the number of fatigue cycles, and the formula could predict the fatigue life of AFS simply. Other scholars have also proposed corresponding fitting formulas based on their own studies. Burman et al. and Waloddi et al. [29,30] used a two-parameter Weibull function to predict fatigue life, and the experimental results showed good fitting results. Kanny et al. [31] gave empirical expressions for the stress and the number of cycles based on experimental data, and this method could simply predict fatigue life of the specimen. The fatigue life prediction curve describes the number of cycles when the specimen fails under different fatigue loading, but it does not reflect the specimen’s damage process during fatigue experiments. Furthermore, most of the fatigue tests are too time-consuming and expensive, so it is necessary to establish the relevant model to describe the damage process of fatigue experiments. Hwang et al. [32] proposed a new concept named the fatigue modulus and applied it to predict fatigue life. The fatigue modulus was defined as the radio of the applied stress to total strain at a specific cycle, and the fatigue modulus degradation rate could be described by an empirical power-law function *An^C^*. When fatigue proceeded to a certain number of cycles, the fatigue modulus could be expressed as:(5)$$G_{f}\left(n\right)=G_{0}-An^{c}$$
where $G_{f}\left(n\right)$ represents the theoretical fatigue modulus value at the *n*th cycle; $G_{0}$ represents the static fatigue modulus value at the initial time; *A* and *c* are both material constants and are determined by the material’s properties. Clark et al. [33] indicated that the fatigue process was divided into two stages, the first stage was the fatigue emergence stage, and the second stage was the fatigue damage stage. The cycle number of the sample’s initial damage was defined as *n_if_*; fatigue damage occurred only in the second stage, and the change of fatigue modulus also occurred in the second stage. Therefore, Clark et al. modified Equation (5) as follows.
(6)$$G_{f}\left(n\right)=\left\{\begin{array}{c} G_{0}\;\;\;\;\;\;\;\;\;\;\;\;\;\;\;\;\;\;n\leq n_{if}\\ G_{0}-Ae^{\left(n-n_{if}\right)c}\;\;n\geq n_{if} \end{array}\right.$$

When the specimen was subjected to the three-point bending load, the overall deflection change of the specimen could be defined as the sum of the bending deflection and shear deflection, i.e.,
(7)$$\delta =\delta_{b}+\delta_{s}=\frac{Fl^{3}}{48{\left(EI\right)}_{eq}}+\frac{Fl}{4{\left(AG\right)}_{eq}}$$
where *F* represents the applied load; *l* represents the span length; ${\left(EI\right)}_{eq}$ and ${\left(AG\right)}_{eq}$ represent the equivalent bending stiffness and equivalent shear stiffness, respectively. The relevant formulas are shown in Equations (8) and (9).
(8)$${\left(EI\right)}_{eq}=\frac{E_{f}btd^{2}}{2}+\frac{E_{f}bt^{3}}{6}+\frac{E_{c}bc^{3}}{12}\approx \frac{E_{f}btd^{2}}{2}$$
(9)$${\left(AG\right)}_{eq}=\frac{bd^{2}}{c}G_{c}\approx bcG_{c}$$

The previous experiments have shown that the failure form of an AFS made by the powder metallurgy method was core shear, so it could be assumed that the bending deflection was not related to the number of cycles, and the shear deflection was related to the number of cycles. Therefore, the fatigue deflection of AFS could be expressed as:(10)$$\delta_{s}=\delta -\delta_{b}$$

According to Equation (7), the fatigue modulus could be expressed as:(11)$$G_{f}\left(n\right)=\frac{Fl}{4bc\delta_{s}}$$

According to Equation (11), the fatigue modulus was obtained by bringing corresponding parameters. The variation curves of the fatigue modulus with the relative cycle number are shown in Figure 10. The relative cycle number could be expressed as the ratio of the cycle number *n* to the failure cycle number *n_f_*.

The fatigue modulus degeneration relation of an AFS made by the powder metallurgy method at 80% and 70% load levels is shown in Figure 10. The overall degradation rude could be divided into three stages. The AB section was the first stage, in the initial stage of the fatigue process, the fatigue modulus of the specimen dropped to a stable value in a very short time. The second section was the BC stage, in which the fatigue modulus remained almost unchanged. This stage occupied 90% of the whole stage. The third section was the CD stage. With the accumulation of the fatigue damage, when the fatigue damage dropped to a critical level, the fatigue modulus dropped abruptly, and the fatigue modulus degraded rapidly in a short number of cycles until the specimen failed.

### 4.2. Fatigue Damage Model

In the general fatigue damage model, the fatigue damage of the material accumulated gradually from the beginning state. Assuming that the frequency and temperature remained constant, the fatigue damage value was 0 when the number of fatigue cycles was 0; the fatigue damage value was 1 when the number of fatigue cycles was equal to the fatigue life. Fatigue damage could be expressed in terms of *D*. The fatigue damage was expressed as follows:(12)$$\begin{array}{c} D=0,\mathrm{where}\;n=0\\ D=1,\mathrm{where}\;n=n_{f} \end{array}$$

In the cumulative damage model, the fatigue damage could be defined in terms of the linear degree of the structural response [33]. When the number of cycles was *n* ≤ *n_if_*, the fatigue damage value was 0, and when the number of cycles was *n* = *n_f_*, the fatigue damage value was 1. The fatigue damage could be expressed as:(13)$$\begin{array}{c} D_{f}\left(n\right)=0,\mathrm{where}\;n\leq n_{if}\\ {0<D}_{f}\left(n\right)\leq 1,\mathrm{where}\;n_{if}<n\leq n_{f} \end{array}$$

The model used in this paper was a degradation damage model based on fatigue modulus, which could be expressed as:(14)$$D_{f}\left(n\right)=1-\frac{G_{f}\left(n\right)}{G_{f}\left(0\right)}$$

According to Equation (14), the variation curves of the fatigue damage with the number of cycles are shown in Figure 11. It can be clearly seen that the fatigue damage gradually increased from 0 to 1 when the number of fatigue cycles grew from 0 to fatigue life, and when the fatigue damage was equal to 1, it was the fatigue failure of the sample. The overall trend of the fatigue damage was similar to the trend of the fatigue modulus. The fatigue damage value changed significantly at the beginning and end of the fatigue process and slowly at the intermediate stage of fatigue damage. Wang et al. [34] used the exponential fitting method to study the fatigue damage fitting method. The selection of the fitting function was related to the features of the material itself and also related to the preparation method, sample size, etc. In this paper, we tried to fit the fatigue damage process with an exponential function, and the fitting results are shown in Figure 12. It can be seen that the effect of the exponential function was not very satisfactory for the imminent failure stage. By comparing the *R*^2^ values in Figure 11 and Figure 12, the *R*^2^ values of the polynomial fitting method are 0.80 and 0.92 when the load levels are 80% and 70%, respectively; while that of the exponential fitting method are 0.75 and 0.87 when the load levels are 80% and 70%, respectively. The polynomial fitting method has a better fitting degree. Therefore, the polynomial fitting method was adopted in this paper, the fitting curves as shown in Figure 11. The fitting curves used the cubic polynomial fitting, the curves could be good fitting for the fatigue damage process. The fitted curves could be expressed as follows.
(15)$$D_{f}\left(n\right)=a+bN+cN^{2}+dN^{3}$$
where *a*, *b*, *c*, and *d* are parameters related to the load levels.

From Figure 7, Figure 11 and Figure 12, it could be seen that the AFS had a large variation of deflection and a large span of fatigue modulus and fatigue damage in the initial stage of bearing fatigue load, and then it would enter a stable stage. In the stable stage, the variations of deflection, fatigue modulus, and fatigue damage floated in a small range, and this part accounted for about 90% of this fatigue process. When the fatigue damage accumulated to a critical level, macroscopic cracks formed on the surface. The cracks expanded rapidly, and the specimen eventually failed under the action of the fatigue load. Therefore, in the fatigue initial stage and failure stage, the deflection, fatigue modulus, and fatigue damage changed in a wide range within a short period of time, and the fitting effect was not ideal. In summary, Figure 13 gives the fatigue damage curves and fitting curves of the stable stage of the fatigue process, and the fitting function was the cubic polynomial, which could be seen that the fitting curves fitted the fatigue damage curves very well.

## 5. Conclusions

In this paper, an aluminum foam sandwich (AFS) was successfully prepared by the rolling composite-powder metallurgy method. The microstructure, fatigue life, deflection curve, failure mode, and fatigue damage of the AFS were studied to evaluate the fatigue performance of an AFS made by the rolling composite-powder metallurgy method (hereinafter referred to as the powder metallurgy method).

Compared with the AFS made by the adhesive method, the AFS made by the powder metallurgy method had a flat bonding interface, no obvious defects, and a high bonding strength. The number of fatigue cycles increased with the decrease of the load level. The number of fatigue cycles of the AFS made by the powder metallurgy method was much larger than that made by the adhesive method. Its deflection-life curve under different load levels was analyzed to study the mechanical variation behavior. It was found that the formation and propagation of the macroscopic cracks accounted for the last 10% of the whole fatigue process, and there was no significant change in the rest stages. The failure modes of the AFSs made by the powder metallurgy method and made by the adhesive method were core shear; and core shear, interface debonding, respectively. The fatigue damage process was also studied, and the fitting degree of different fitting methods was explored. When the load level was 70%, the R^2^ values of the polynomial fitting method and the exponential fitting method were 0.92 and 0.87, respectively. The polynomial fitting method had a better fitting degree and the best fitting effect.

In conclusion: we used a systematic method to investigate the fatigue properties of an AFS made by the rolling composite-powder metallurgy method, and the obtained results are of great significance for evaluating the properties and promoting the application of AFSs made by the rolling composite-powder metallurgy method.

## Figures and Tables

**Figure 1 materials-16-01226-f001:**
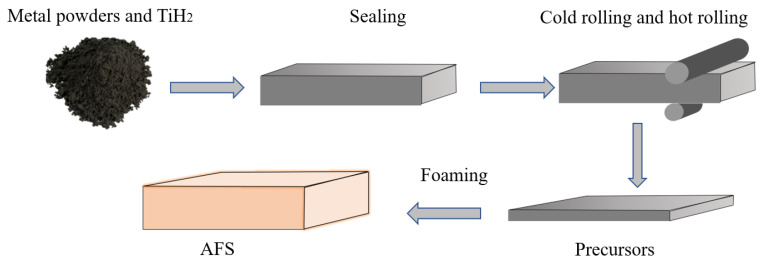
Process flow chart of an AFS made by the powder metallurgy method.

**Figure 2 materials-16-01226-f002:**
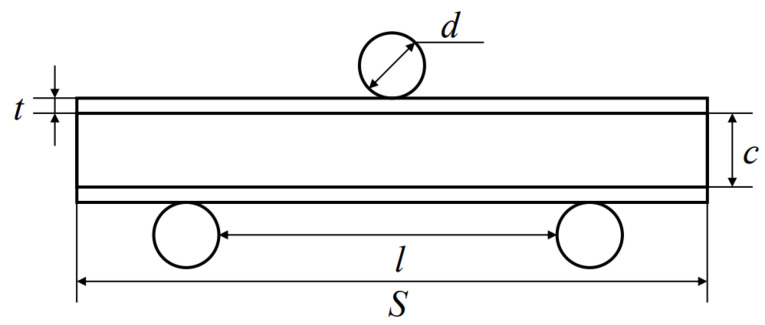
The position relationship between the three-point bending fixture and the sample.

**Figure 3 materials-16-01226-f003:**
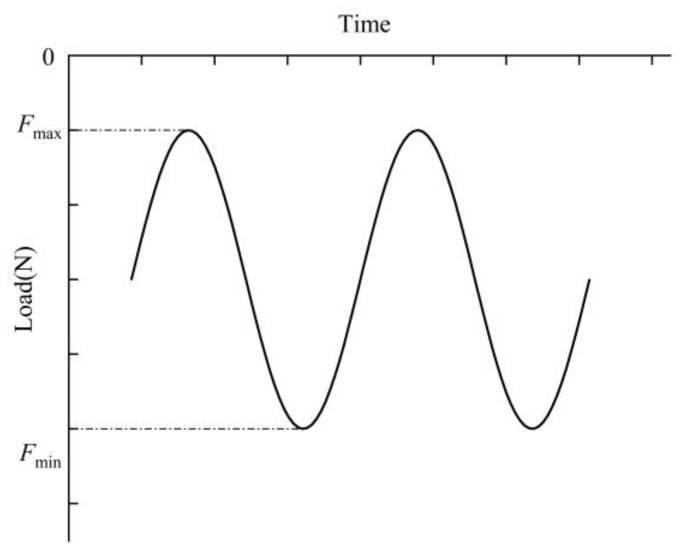
The fatigue load curve of the three-point bending test.

**Figure 4 materials-16-01226-f004:**
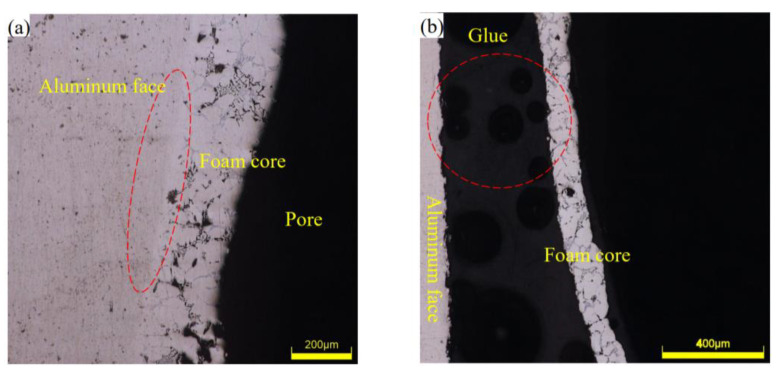
Microscopic interface: (**a**) the powder metallurgy method, (**b**) the adhesive method.

**Figure 5 materials-16-01226-f005:**
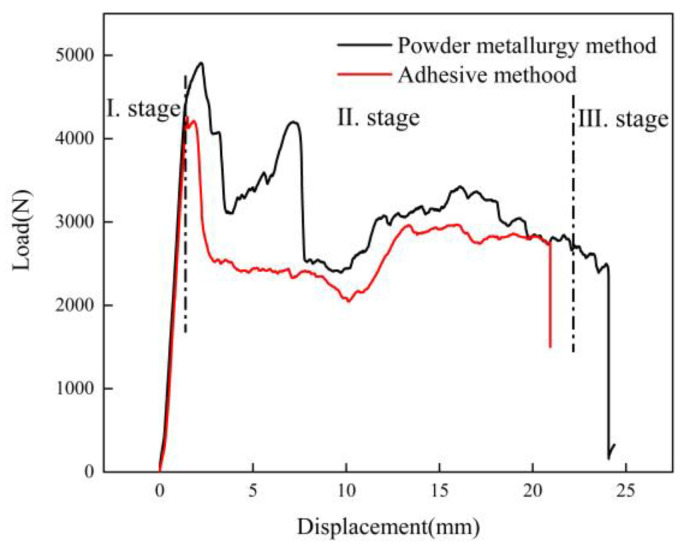
Load-displacement curves of an AFS made by different preparation methods.

**Figure 6 materials-16-01226-f006:**
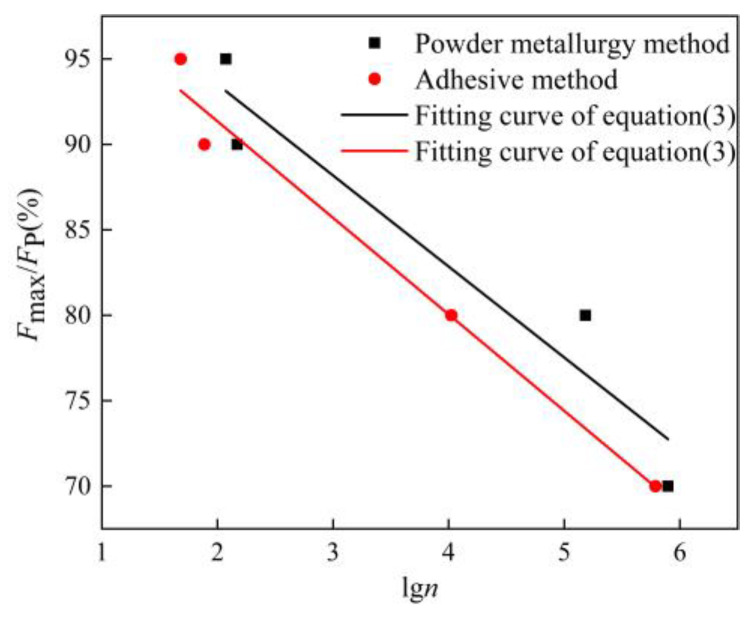
Fatigue life curves of AFSs made by different preparation methods.

**Figure 7 materials-16-01226-f007:**
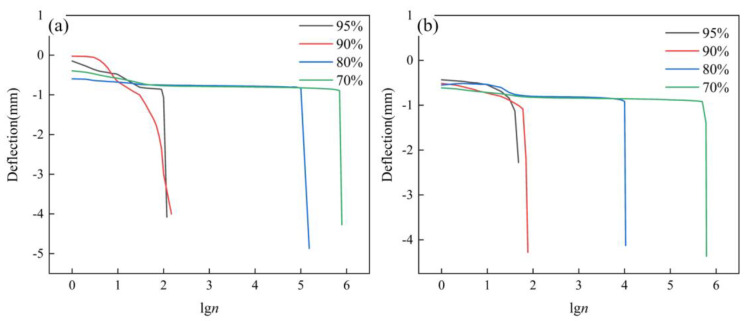
Deflection life curves at different load levels. (**a**) AFS prepared by the powder metallurgy method, (**b**) AFS prepared by the adhesive method.

**Figure 8 materials-16-01226-f008:**
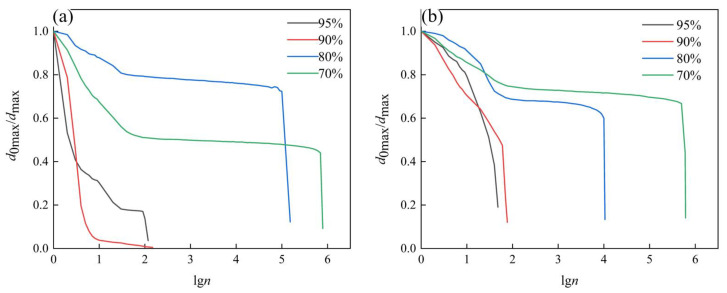
Relative displacement curves at different load levels. (**a**) AFS prepared by the powder metallurgy method, (**b**) AFS prepared by the adhesive method.

**Figure 9 materials-16-01226-f009:**
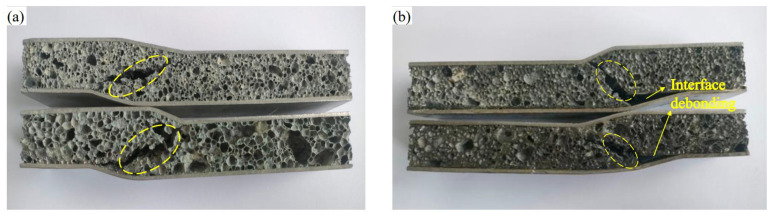
Fatigue fracture diagram of AFSs: (**a**) the powder metallurgy method, (**b**) the adhesive method.

**Figure 10 materials-16-01226-f010:**
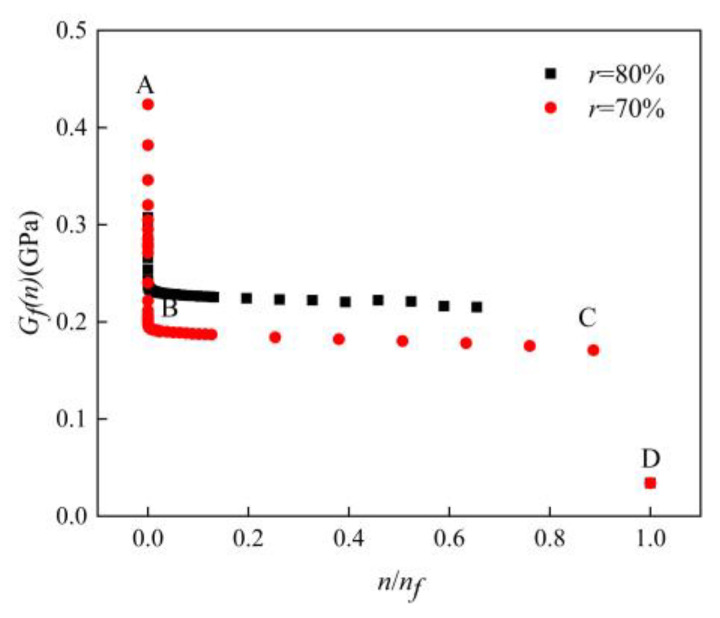
Fatigue modulus variation curves. (Specimens made by the powder metallurgy method).

**Figure 11 materials-16-01226-f011:**
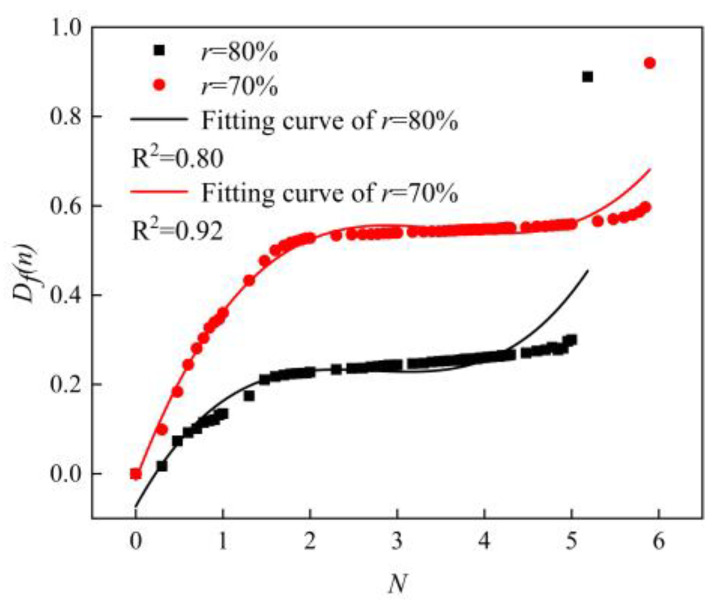
Fatigue damage curves. (Specimens made by the powder metallurgy method, *N* = lg*n*).

**Figure 12 materials-16-01226-f012:**
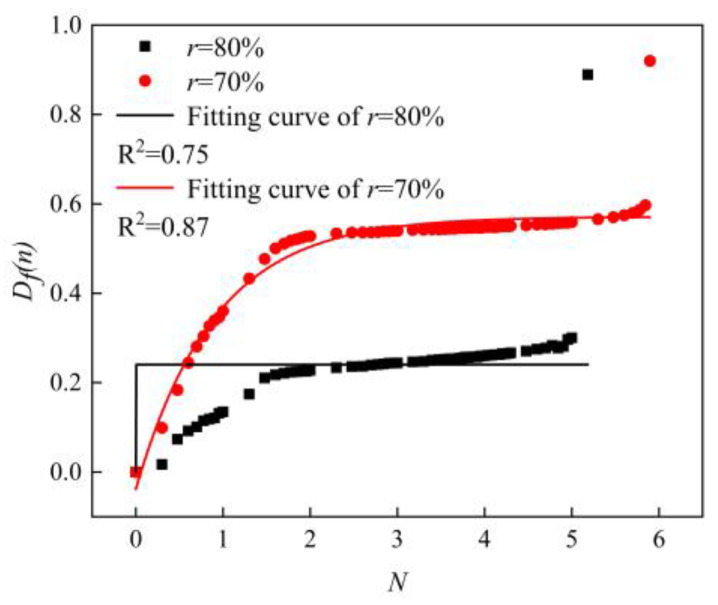
Exponential fitting results. (Specimens made by the powder metallurgy method, *N* = lg*n*).

**Figure 13 materials-16-01226-f013:**
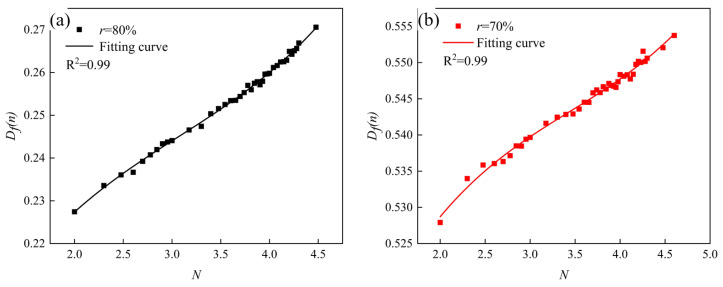
Fatigue damage fitting curves at the stable stage. (Specimens made by the powder metallurgy method, *N* = lg*n*). (**a**) *r* = 80%, (**b**) *r* = 70%.

**Table 1 materials-16-01226-t001:** Elemental composition of mixed powders [26].

Composition	Range Size (μm)	Purity (%)	Content
Al	<45	99.7	85%
Si	<38	99.5	6%
Mg	<75	99.9	4%
Cu	<38	99.9	4%
TiH_2_	<45	99.7	1%

**Table 2 materials-16-01226-t002:** Fixture size.

	*S*	*l*	*d*
Size (mm)	170	100	10

**Table 3 materials-16-01226-t003:** Fatigue life of AFSs made by different preparation methods.

$F_{max}$$/F_{P}$ (%)	95	90	80	70
*n* _Adhesive method_	48	77	10,541	615,201
*n* _Powder metallurgy method_	118	148	152,730	789,508

## Data Availability

Not applicable.

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
