# Peer review of "Fatigue of an Aluminum Foam Sandwich Formed by Powder Metallurgy"

_materials, 2023, doi:10.3390/ma16031226_

Round 1
Reviewer 1 Report
In this manuscript, the authors show ‘Study on fatigue properties of aluminum foam sandwich panels by rolling composite-powder metallurgy methods’. They present nice spectrum of microstructure and mechanical. The results are very interesting and the paper is generally well-prepared. However, in my opinion, there are some points that authors should change to improve the paper's quality. During its careful review, a series of problems and suggestions (listed below) arose.
My recommendation is Major revision.
Comments
1. Please mention the detail of raw materials (Al, Si, Mg and Cu) like average particle size, particle size distribution curve and SEM image of particles.
2. In section 2.3, please mention the room temperature in three-point bending test conditions. How many samples are used in the three-point bending test?
3. Mention the error bar in the figure and improve the figure image.
4. Show the load-deflection curve of both the sandwich structure of 3-point bending and also explain the flexural deformation behavior of the panel with the mechanism
5. Compare the fatigue life of the sandwich panels with the bare core foam samples and also compare the failure mechanism.
6. Also show the effect of different face sheet thickness on fatigue life of both type of sandwich structure and bare foam should be explained.
7. English expression needs to be improved. The manuscript contains some grammatical errors and unreasonable sentence structure.
Author Response
Comment 1: Please mention the detail of raw materials (Al, Si, Mg and Cu) like average particle size, particle size distribution curve and SEM image of particles. (in red)
Response: We are so grateful for your kind question. The range size, purity, and content of the metal powders and blowing agent TiH2 have been added to the paper using the table (Line 104, Table 1). The raw material is common commercial elements.
Comment 2: In section 2.3, please mention the room temperature in three-point bending test conditions. How many samples are used in the three-point bending test? (in red)
Response: We are so grateful for your kind question. The relevant content has been added in section 2.3 that you suggested (Line 129).
Comment 3: Mention the error bar in the figure and improve the figure image. (in red)
Response: We are so grateful for your kind question. Your suggestion is to the add error bar in the figure. So we have added R2 values in Figure 12, 13, 14 (Line 385, 387, 401) to evaluate the fitting degree, and other figures were directly obtained from the experimental results.
Comment 4: Show the load-deflection curve of both the sandwich structure of 3-point bending and also explain the flexural deformation behavior of the panel with the mechanism.
Response: We are so grateful for your kind question. Thank you very much for your advice. This opinion is of a great value to the study that AFS is subjected to the quasi-static three-point bending experiments. This analysis process was shown in Section 3.2 (Line 173). However, the objective of this paper is to explore the fatigue performance of AFS made by different combination methods. Around this objective, we explored the fatigue life, deflection curve, and fatigue damage and obtained better results. The current data in this paper is sufficient to explore the fatigue properties of AFS with different preparation methods. (Wang, L.; Zhang, Y. W.; Ho, J. C. M.; Lai, M. H., Fatigue behaviour of composite sandwich beams strengthened with GFRP stiffeners. Engineering Structures, 2020, 214.) Thanks again for your advice, which is necessary for our subsequent research work.
Comment 5: Compare the fatigue life of the sandwich panels with the bare core foam samples and also compare the failure mechanism.
Response: We are so grateful for your kind question. Your suggestion is very helpful to study the influence of different factors on the fatigue life of AFS. However, the research in this paper is to compare the fatigue life of AFS with different preparation methods and to analyze the influence of preparation methods on the fatigue life of AFS. The current data in this paper is sufficient to explore the fatigue properties of AFS with different preparation methods. (Yao, C.; Hu, Z.; Mo, F., Three-Point Bending Fatigue Behavior of Aluminum Foam Sandwich Panels with Different Density Core Material. Metals, 2021, 11, (10), 8-11.) Thanks again for your advice, which is necessary for our subsequent research work.
Comment 6: Also show the effect of different face sheet thickness on fatigue life of both type of sandwich structure and bare foam should be explained.
Response: We are so grateful for your kind question. Your suggestion is very helpful to study the influence of different thicknesses on the fatigue life of AFS. However, the research in this paper is to compare the fatigue life of AFS with different preparation methods and to analyze the influence of preparation methods on the fatigue life of AFS. The current data in this paper is sufficient to explore the fatigue properties of AFS with different preparation methods. (Yao, C.; Hu, Z.; Mo, F., Three-Point Bending Fatigue Behavior of Aluminum Foam Sandwich Panels with Different Density Core Material. Metals, 2021, 11, (10), 8-11.) Thanks again for your advice, which is necessary for our subsequent research work. In the future, we will study the fatigue properties of AFS prepared by the rolling composite-powder metallurgy method with different thicknesses.
Comment 7: English expression needs to be improved. The manuscript contains some grammatical errors and unreasonable sentence structure.
Response: We regret there were problems with the English. The paper has been carefully revised by Dr. Huang to improve the grammar and readability.

Reviewer 2 Report
The article "Study on fatigue properties of aluminum foam sandwich panels by rolling composite-powder metallurgy methods" discusses the fabrication of aluminum foam sandwich panels, conduction of fatigue test and analysis of properties. This is well organized and fit into the scope of the journal. At the same time, there are many points have to be improved to enhance the quality of the article.
1. A thorough language correction / checking is needed. Many grammatical mistakes were noticed.
1. Introduction
2. The authors are very much reluctant in the review of literature. (Ex: Zhang et al. [10-12] studied – Whether the references 10, 11 & 12 were authored by Zhang et al.? Elnasri et al. [13-16] studied – Are the references 13, 14, 15 & 16 authored by Elnasri et al.?). So the entire section should be corrected with the correct references.
2.1. Preparation of materials
3. The temperature was 470℃ and the time was 1.5h. How these parameters were decided?
4. 3003 aluminum alloy – can be written as aluminum alloy 3003
5. the cavity was cleaned and immersed in 40 g/L 85 aqueous sodium hydroxide solution for 10-20 min and then in dilute hydrochloric acid 86 solution for 10 min to remove the impurities on the surface – Include the proof for the parameters used.
6. the panel was 100 bonded with the aluminum foam core layer and then put into the drying box at 80℃ for 101 2h and could be used after resting for 48h. – Provide the proof for the parameters used.
2.3. Fatigue test
7. Equations can be written by using recommended equation editor and should be numbered as per journal guidelines.
8. What is Fp?
9. What are ‘t’ & ‘c’ in figure 1?
10. Figure 1. The schematic diagram of the experimental device. – Is it experimental device or fixture?
11. What are the values in X and Y axis in Figure 2. The fatigue load curve of three-point bending test? How it was made? What are the inference?
12. Figure 3. Schematic diagram of the fatigue test – Is it Schematic diagram or experimental setup?
3.1. Microstructure
13. How do you check the Microstructure? And where it was done?
3.5. Failure modes
14. How do you obtained Figure 9. Fatigue fracture diagram of AFS?
15. What is the inference of red color markings in figure 9 (a)
4.1. Fatigue modulus
16. In the above paper, we proposed formulas – What it means?
17. Burman et al. [28,29] – Reference 28 & 29 are not same. Use a correct one.
18. Hwang et al. [31] proposed (line 290) / Clark et al. [31] showed (line 298) - Which one is correct? – This shows the authors are not reviewing the articles referred.
19. What about the fatigue modulus variation curves for other specimens?
4.2. Fatigue damage model
20. What about the fatigue damage curves for other specimens?
Reviewer 3 Report
I have the following comments on the submitted article:
1. Line 77, first sentence, insert a literary source for the production of powders by the method of powder metallurgy, e.g. Studeny, Z.; Krbata, M.; Dobrocky, D.; Eckert, M.; Ciger, R.; Kohutiar, M.; Mikus, P. Analysis of Tribological Properties of Powdered Tool Steels M390 and M398 in Contact with Al2O3. Materials 2022, 15, 7562. https://doi.org/10.3390/ma15217562
2. Line 121: What is the value of Fp-ultimate bending strength?
3. Table 1. center the values in the table.
4. Fig. 3 enlarge the image and insert a description of the device (arrows)
5. Fig. 4 I will consider the images, enlarge the letters in the images and possibly change the font color to yellow for greater contrast.
6. Figure 5. slightly enlarge the image and insert curves into the image that will depict the three stages described in the text above the image.
7. Table 2 shows load levels 95, 90, 80, 70%, but in line 121-122 you state 90, 80, 70 and 65%. It is probably necessary to rewrite it in lines 121 - 122.
8. I will ask somewhere at the end of paragraph 3.3 to insert a sentence about the comparison of life in percentages.
9. Fig. 6. the description on the Y-line axis is not readable, fix it (badly cropped graph)
10. Fig. 8a. explain in the text why the blue curve of 80% in the given picture reaches higher values of relative displacement than the green curve of 70%.
11. Fig. 9. change the color of the text and the dashed bounded area to yellow for better contrast.
12. Fig. 9. according to these pictures, doesn't figure 9a show a greater porosity of the material than figure 9b?
13. Insert Figures 11, 12, 13 into the graph of coefficient of determination R2 values
Conclusion:
No. 2: insert a percentage comparison of the results of fatigue cycles.
no. 3: the same as in no. 2
insert a conclusion that will be devoted to the comparison of the coefficient of determination R2 and the measured values.
summary:
The authors wrote a high-quality article in which they analyzed the obtained results using a scientific approach. After correcting my comments, I will recommend the article for pubicizing.
Highlight the supplemented and modified text in yellow.
Well thank you.
Reviewer 4 Report
Overall comments (please refer to the uploaded reviewed paper for in depth suggestions):
1. The reviewer believed that the current research work based on the aluminum foam sandwich (AFM) developed via powder metallurgy method has been extensively studied. The reviewer failed to identify the uniqueness/novelty of the current study in comparison with the existing studies.
2.The paper needs to be extensively proofread as the content of the paper is quite challenging to be understood in addition to the poor flow of the paper.
3. It is highly recommended to include/study the effects of various porosity percentage (varying the content of TiH4) on the AFS properties investigated instead of focusing on one porosity percentage so that an important novelty could be highlighted for the current study.
3. It is highly recommended to summarize the important/critical findings of the previous studies as stated in the introduction section, paragraph 2 instead of mentioning the studies.
4. The term used which is 'large holes' as in the introduction section (paragraph 2) is quite confusing and needs to be properly explained. The reviewer believed that that term is referring to the pores in the foam. Please recheck and use the correct term.
5. It is highly recommended to include a schematic diagram/illustration to describe the steps involved in the AFS fabrication via powder metallurgy technique.
6. The findings for microstructure of AFS were poorly discussed with unclear explanation that related to the microscopic analysis (Figure 4) (Please refer to the uploaded reviewed paper).
7. The finding on the ultimate strength is believed cannot be obtained from the load-displacement curve. It is highly recommended to include the stress strain curve to obtain the said strength. Please recheck and make necessary improvements on this part.
3.The terms used in the discussion part under failure modes (section 3.5) need to be carefully selected as the authors used the terms 'bubble holes' to describe the cracking phenomenon in which the reviewer believed that the authors refer to the pores created by the blowing agent of TiH4. It is highly recommended to recheck and use the appropriate term accordingly.
8. The conclusion should be improved by summarizing important/critical findings.
Round 2
Reviewer 1 Report
The article reached a form that is now suitable for publication.
Author Response
Thank you very much for your reply. We really appreciate your support and recognition of our work. Thank you again for your efforts.
Reviewer 2 Report
All the suggestions were addressed by the authors. Hence it may be accepted
Author Response

(The authors gave the same response as above.)

Reviewer 3 Report
Thanks to the authors for completing my comments. I still have these small reservations about the article:
1. Line 126 - enter the equation on one line.
2. Figure 4. remove (this is not a schematic diagram) it is an experimental device. Figure 2 is enough to understand.
3. Figure 6. dotted lines (change their color) now it looks like they belong to the adhesive method (since it is also red). Mark as follows: I. stage; II. stage; III. stage
4. Figure 11. enlarge (or at least enlarge the labels on the axes)
5 Enlarge images 12 and 13 slightly (very small font in the graphs)
Please mark the corrected text in yellow.
That's all, just small form errors.
Reviewer 4 Report
Dear authors,
1. Thank you very much for responding each of my comments. My main issue is actually regarding the novelty of the paper. I do understand that the authors actually comparing two (2) different techniques between powder metallurgy and adhesive. However, I have found extensive works on both techniques involving Al foam in which the pores were generated from the decomposition of TiH2 space holder. In view of this, I had suggested to add additional results on the porosity percentage so that the novelty of the current work could be justified.
2. The microstructure evaluation should be included in the materials and methods part prior to be discussed in the discussion part so that readers are aware about the type of machine used and its preparation involved before obtaining the relevant microstructure.
3. The authors did mention about the fatigue mechanism in the introduction part. Is that means the authors should provide any relevant reaction or perhaps equation during the fatigue failure?
Round 3
Reviewer 3 Report
Thanks for all the corrections. The article can be published in this form.
Author Response

(The authors gave the same response as above.)

Reviewer 4 Report
Dear authors,
Thank you for replying to my comments. I do concern about the novelty of the paper and the content of the paper as well. You may resubmit after the previous concerns had been addressed accordingly. Perhaps, you may submit the current paper to another potential/suitable journal after proofreading.
Thank you very much.
Author Response
We appreciate your comments. There are many methods to prepare AFS, but most of them are limited to the laboratory, can not produce large-size AFS, and the interface bonding strength is low, and powder density is not high. Based on the existing problems, we put forward the method of the rolling composite-powder metallurgy method to prepare AFS. This method is a kind of manufacturing process that can be used to prepare large-size AFS in industry. This method adopts the powder metallurgy technology route, which not only can form a metallurgical bonding interface but also has a controllable preparation process. Therefore, it has great development prospect. We plan to apply this technology to aerospace applications, such as aircraft flooring. We have carried out the research on engineering applications with relevant manufacturers. In this process, we found that the fatigue performance of aircraft components is an important parameter to measure the performance of aircraft components, so it is of great significance to study the fatigue performance of AFS made by the rolling composite-powder metallurgy method. In order to measure the fatigue performance of AFS, we chose AFS made by adhesive method to compare with it and used a systematic method to compare the fatigue performance. The final results show that the fatigue performance of AFS prepared by the rolling composite-powder metallurgy method is better than that of the adhesive method, and the fatigue life prediction formula is obtained to predict the fatigue life of AFS. This result is of great significance for our future application of AFS in the aerospace field.